# The Use of Coffee Cherry Pulp Extract as an Alternative to an Antibiotic Growth Promoter in Broiler Diets

**DOI:** 10.3390/ani15020244

**Published:** 2025-01-16

**Authors:** Wanaporn Tapingkae, Phatchari Srinual, Pimporn Khamtavee, Naret Pintalerd, Thanongsak Chaiyaso, Mongkol Yachai, Chanidapha Kanmanee, Chompunut Lumsangkul, Orranee Srinual

**Affiliations:** 1Department of Animal and Aquatic Sciences, Faculty of Agriculture, Chiang Mai University, Chiang Mai 50200, Thailand; wanaporn.t@cmu.ac.th (W.T.); phatchari_sri@cmu.ac.th (P.S.); pimporn.k@cmu.ac.th (P.K.); chanidapha.k@cmu.ac.th (C.K.); 2Functional Feed Innovation Center (FuncFeed), Faculty of Agriculture, Chiang Mai University, Chiang Mai 50200, Thailand; 3Highland Research and Training Center, Faculty of Agriculture, Chiang Mai University, Chiang Mai 50200, Thailand; naret.p@cmu.ac.th; 4Division of Biotechnology, Faculty of Agro-Industry, Chiang Mai University, Chiang Mai 50100, Thailand; thanongsak.c@cmu.ac.th; 5Faculty of Animal Science and Technology, Maejo University, Chiang Mai 50290, Thailand; mongkol_yc@mju.ac.th; 6Office of Research Administration, Chiang Mai University, Chiang Mai 50200, Thailand; 7Department of Animal Science, National Chung Hsing University, Taichung 40227, Taiwan; lumsangkul@nchu.edu.tw

**Keywords:** antibiotic, antioxidant, broiler chicken, coffee pulp, sustainability

## Abstract

This study investigated the effect of different levels of coffee cherry pulp extract (CCPE) added to broilers’ diet on growth performance, meat quality, carcass characteristics, and health parameters. The results indicate that dietary CCPE supplementation positively affects broilers’ growth performance and health. Moreover, CCPE is a feed additive that can replace antibiotics in broiler feed.

## 1. Introduction

Coffee production has achieved substantial increases and is expected to continue this rising trend. Thailand is the third-largest coffee producer in Asia [1,2]. Overall, 11,000 tons of Arabica coffee are produced in North Asia. During the processing of green beans, the skin, pulp, mucilage, and parchment are discarded as agricultural by-products, making up 55% of the total coffee [2]. Therefore, the coffee production process produces many by-products, constituting waste that harms the environment. Coffee cherry pulp (CCP) is the by-product of the wet process. For every two tons of fresh cherry coffee that are processed, one ton of coffee pulp is produced [3,4]. CCP constitutes 40–50% of the coffee fruit’s weight and contains 4–12 g protein, 1–2 g lipids, and 45–89 g carbohydrates per 100 g of dry matter. It also includes compounds like pectin, tannins, and caffeine (0.12–0.26%), as well as phenolic acids such as caffeic, gallic, and chlorogenic acids [5]. Moreover, CCP has a high content of bioactive compounds, especially polyphenols, which have antioxidants, anti-inflammatory, antimicrobial, antiallergic, and anticancer activities [3,6]. Therefore, the extracts from CCP are a source of phytochemicals, including chlorogenic acids (CGA) and caffeine, which contribute to its beneficial properties [7]. Machado et al. [8] studied the chemical composition and bioactive potential of CCP and the results showed that CCP has CGA (220.56 mg/100 g) and caffeine (0.85 g/100 g) content. CGA is a well-known phenolic compound with antioxidant properties, while caffeine contributes to metabolic stimulation and other physiological effects [9,10].

These bioactive compounds can enhance animal health; for example, CGA improved growth performance, immunity, antioxidant status, and intestinal barrier function [11,12], alleviated oxidative stress and ameliorated hepatic inflammation [13] in broilers, and also improved gut health and growth performance in weaned piglets [14]. Moreover, caffeine supplementation increased body weight gain, decreased the feed conversion ratio (FCR), and improved blood biochemistry in broilers [15]. There have been many studies on the use of CCP in animal feed. For instance, it can substitute up to 20% of conventional feed for livestock, including pigs and poultry, providing essential nutrients while lowering feed costs [4,16]. Donkoh et al. [17] found that incorporating up to 2.5% CCP in chicken diets for 8 weeks was acceptable, while higher levels reduced weight gain. Recent studies by Antúnez et al. [18] have shown that substituting up to 25% of wheat bran with extruded coffee pulp flour in broiler diets does not negatively impact performance and can improve certain performance metrics while maintaining intestinal health.

However, there is currently no research that explores the use of coffee cherry pulp extract (CCPE) as a dietary supplement in animal feed. This gap in the literature has prompted researchers to investigate the potential benefits of using CCPE in broiler chicken diets. This study aimed to evaluate its effects of CCPE on growth performance, serum biochemistry, carcass characteristics, meat quality, intestinal morphology, gut microbiota, and gene expression in broiler chickens. Thus, the addition of feed additives is a strategy to reduce animal feed costs. It can also improve feed efficiency in animals and replace the use of antibiotics in livestock farming.

## 2. Materials and Methods

### 2.1. Coffee Cherry Pulp Preparation and Extraction

CCP was obtained through the wet processing method, which involved mechanically peeling freshly harvested ripe coffee fruits from the Highland Research and Training Center at Chiang Mai University. To enhance shelf life and reduce water activity, the CCP was dried at 60 °C for 48 h. Subsequently, a coffee grinder was used to grind the dried pulp into a uniform powder with a particle size of 0.1 mm. Following the methodology outlined by Gligor et al. [19], a solvent of 95% ethanol (Merck, Darmstadt, Germany) was used, employing a solid-to-liquid ratio of 1:5 g/mL. Specifically, 10 g of CCP was accurately weighed and combined with 50 mL of 95% ethanol within a Soxhlet extractor thimble, which was subsequently placed in the extraction apparatus. The temperature of the heating plate was carefully set to 90 °C, following the procedures established by Kusuma et al. [20], and the extraction was conducted for 60 min. Following extraction, the CCPE was concentrated using a rotary vacuum evaporator at 60 °C, as described by Vijayalaxmi et al. [21]. The components of the CCPE utilized in the experiment are presented in Table 1.

### 2.2. Chicks, Diets, and Experimental Design

The present study was approved by the Department of Animal and Aquatic Animal Sciences at the Faculty of Agriculture, Chiang Mai University, Thailand. All experimental procedures were conducted in accordance with the guidelines established by the Local Experimental Animal Care Committee. The ethical approval code for this study was AG02008/2566. A total of 500 one-day-old Ross 308 male chicks with similar initial weights were randomly assigned to five treatment groups, with 10 experimental cages per treatment (10 birds per cage). The chicks were given *ad libitum* access to water and feed throughout the experiment. The chicks were fed two distinct diets: a basal starter diet for the first 14 days and a finisher diet for 15 to 35 days. The experimental groups were defined as follows: one group received a basal diet without additives, while another group was administered the basal diet supplemented with an antibiotic growth promoter (AGP), which consisted of a mixture of amoxicillin (100 g/kg) and colistin (400 × 106 IU/kg) at a dosage of 0.25 g/kg [22]. Additionally, three groups were supplemented with 0.5, 1.0, and 2.0 g/kg of CCPE. Table 2 presents the ingredients of the basal diets, which were subjected to proximate analysis according to the methods outlined by AOAC [23]. The duration of the experiment was 35 days, corresponding to the slaughter age of the chicks.

### 2.3. Growth Performance Determination and Economic Evaluation

The body weight (BW) of each chick in each cage (replicate) was recorded at the start and end of the experiment. The amount of feed eaten each day was recorded, allowing for the calculation of key performance indicators, including the average daily feed intake (ADFI), average daily gain (ADG), and feed conversion ratio (FCR). Feed intake (FI) was calculated by subtracting the amount of feed remaining from the amount of feed provided on each feeding day. BWG = BW end of the experiment—BW start of the experiment; ADG = BWG/length of the period; ADFI = FI/length of the period; FCR = ADFI/length of the period. The economic analysis was based on feed costs (0.57 USD/kg), feed intake, FCR and BW gain during the experimental period. An economic analysis was conducted to evaluate the total feed cost and revenue, using the farm-gate price of meat (1.14 USD/kg) as described by Son et al. [24]. The economic evaluation was conducted using US dollars (USD) based on the exchange rate information from the Bank of Thailand.Total feed cost = total feed intake per chick × the cost of one kg of feed(1)Total revenue = live body weight × price/kg(2)Feed cost/kg BW gain = FCR × the cost of one kg feed(3)Net profit = total revenue − total feed cost(4)Benefit/cost ratio = total revenue/total feed cost(5)

### 2.4. Serum Biochemistry and Lipid Profiles

Serum biochemistry was collected from 10 chicks per group at 35 days of life. For the analysis of each sample, three replications were performed to ensure the accuracy and reliability of the results. Blood samples were obtained from the wing vein and transferred into sterile tubes without anticoagulant. The serum was separated by centrifugation at 3000 rpm for 15 min and subsequently stored at −20 °C in a refrigerator for subsequent analysis [25]. The serum samples were subjected to a comprehensive analysis of various biochemical parameters, including aspartate aminotransferase (AST), blood urea nitrogen (BUN), creatinine, alanine transaminase (ALT), alkaline phosphatase (ALP), total protein, albumin, and globulin levels. Additionally, a lipid profile assessment was conducted, which included measurements of high-density lipoprotein (HDL) cholesterol, low-density lipoprotein (LDL) cholesterol, total cholesterol (TC), and triglycerides (TGs). The methodologies employed for the analysis of both lipid profiles and serum biochemistry followed the guidelines established by Ding et al. [25] and Ashour et al. [26] Analyses were conducted using commercially available kits (DiaSys Diagnostic Systems GmbH; Holzheim, Germany), and the results were processed with the Automated Chemistry Analyzer, BX-3010.

### 2.5. Characterization of Carcass and Meat Quality

After slaughter, the weights of carcasses (including neck, head, wings, drumsticks, shanks, and skeleton), meat (consisting of breast and thigh), and internal organs (such as spleen, gizzard, liver, abdominal fat, heart, proventriculus, small intestine, and cecum) were recorded and expressed as percentages relative to the broilers’ live weight. Then, the breast and thigh from 10 chicks per group were immediately packed individually in sealable plastic bags and stored at 4 °C for the measurement of meat quality [25]. The pH values were recorded at 15 min, 24 h, and 48 h using a portable pH meter. The drip loss value of the breast was measured at 24 and 48 h. Two grams of breast samples was collected, placed in a sealed plastic bag, and stored at 4 °C for 24 and 48 h. The colors of the breast muscles were measured using a Minolta Chroma Meter, Model CR-410, Minolta Camera Co., Ltd., Osaka, Japan. Before use, the instrument was calibrated with white and blank references. The sample was averaged for L* (lightness), a* (redness), and b* (yellowness) values [27].

Cooking loss and shear force were determined following the protocol described by Qu et al. [28]. Meat samples weighing between 50 and 60 g were prepared with two replicates per sample (10 samples per group). Initially, excess surface moisture was absorbed, and the initial weight was recorded. Each sample was then placed in a heat-resistant bag and sealed securely. Subsequently, the samples were submerged in a water bath at 80 °C for 20 min to ensure the core temperature of the meat reached 70 °C. After removal from the water bath, samples were allowed to cool at room temperature for approximately 30–35 min. Upon cooling, samples were removed from the bag, excess released juices were absorbed, and the final weight after cooking was recorded. Shear force analysis involved cutting the cooked meat samples into 1.27 cm thick squares and measuring the cutting force using a mechanical device (Instron Model 3433 Universal test machine, Norwood, MA, USA). Each sample was subjected to this procedure in triplicate.

### 2.6. Measurement of Intestinal Histomorphology

During chick slaughter, segments were collected from 10 chicks per group, with samples taken from the duodenum, jejunum, and ileum, representing distinct regions of the small intestine. The intestines were rinsed with phosphate-buffered saline (PBS) at pH 7.4 and subsequently fixed in 10% buffered formalin for 24 h. Following fixation, intestinal tissues were embedded in paraffin, sectioned at a thickness of 4 µm, and stained using hematoxylin and eosin (H&E) staining according to Srinual et al. [29]. Histological sections were examined using a compound microscope (A1 Zeiss Axio Scope, Gottingen, Germany) at a magnification of 10x. Morphometric analysis of the small intestine was conducted using a high-quality digital camera (Canon EOS-6D mark, Canon USA, Inc., Huntington, NY, USA) to measure and visualize the morphometric characteristics. Subsequently, morphometric parameters such as villus height (VH), villus width (VW), crypt depth (CD), and villus height to crypt depth ratio (VH:CD) were measured using Motic Images Plus 2.0 software (Motic China Group Co., Xiamen, China).

### 2.7. Composition of Microflora in Cecal Contents

The samples of cecum fluid were collected from 10 chicks per group in sterile tubes. The samples were stored at −20 °C until microbial analysis. One gram of cecal content was serially diluted 10-fold and homogenized with 7 mL of brain heart infusion broth (BHI, Oxoid Ltd., Basingstoke, UK) supplemented with 2 mL of glycerol and then frozen at −20 °C until subsequent analysis following the established protocols of Lefter et al. [30]. Following thawing, the samples were diluted in phosphate-buffered saline at pH 7.4 (PBS, Oxoid Ltd., Basingstoke, UK), and microbial counts were determined as follows: *Lactobacillus* spp. were enumerated using Man, Rogosa, and Sharpe agar selective medium (Oxoid CM0361, Oxoid Ltd., Basingstoke, UK); *Salmonella* spp. were evaluated on *Salmonella–Shigella* agar (Oxoid CM0099, Oxoid Ltd., Basingstoke, UK); and *Escherichia coli* (beta-hemolytic) counts were determined on sheep blood agar (Trypticase soy agar (TSA) 5% *w*/*v*), followed by incubation at 37 °C for 24 h under aerobic conditions. Each sample was subjected to this procedure in triplicate. The results were reported as log10 CFU/g of cecal content. The average outcomes of each microbiological assay were conducted in triplicate and subjected to subsequent statistical analysis.

### 2.8. Assessment of Gene Expressions Related to Immune Response and Antioxidant Activity

One broiler per replicate (10 samples per group) was randomly selected for collecting liver and jejunum tissues. The liver and jejunum tissues were promptly excised and cryopreserved at −20 °C until RNA extraction. Each sample, weighing 50 mg, was homogenized with lysis buffer using a tissue homogenizer. The RNA extraction process was performed following the instructions provided by the manufacturer, using a column-based RNA extraction kit (Invitrogen PureLinkTM RNA Mini Kit). The concentration and purity of total RNA were assessed by measuring the absorbance ratio of 260–280 nm using a NanoDrop 2000 spectrophotometer (Thermo Fisher Scientific, Waltham, MA, USA).

cDNA was generated using the Bio-Rad iScriptTM RT Supermix cDNA synthesis kit (Bio-Rad, Hercules, CA, USA). According to Table 3, the primers required to amplify the genes IL-1β, IL-6, TNF-α, MnSOD, CAT, and GSH-Px1 were generated. The qPCR was conducted using the CFX ConnectTM Real-Time PCR System (BIO-RAD, Hercules, CA, USA) and the iTaq Universal SYBR Green supermix 2X (BIO-RAD, Hercules, CA, USA), along with customized primers for individual genes. The expression levels of antioxidant- and immune-related genes were quantified using the 2^−ΔΔCt^ technique using a standard curve.

### 2.9. Statistical Analysis

All data were analyzed using one-way analysis of variance (ANOVA) and Duncan’s multiple range test for multiple comparisons in SPSS version 23.0 (SPSS Inc., Chicago, IL, USA). The results are reported as means ± standard errors, and differences were considered significant at *p* < 0.05.

## 3. Results

### 3.1. Growth Performance

As presented in Table 4, during the starter phase (1–14 days), broilers receiving CCPE supplementation at all levels exhibited increased ADG compared to the control groups. Additionally, FCR was notably lower in the group supplemented with CCPE at 0.5 and 1.0 g/kg diet, although this difference was not statistically significant in the group supplemented with CCPE at 2.0 g/kg of diet (*p* < 0.05). ADFI was not significantly different (*p* > 0.05). In the final phase (days 15–35), the group receiving 0.5 g/kg of CCPE had the highest ADG, while the 1.0 and 2.0 g/kg CCPE groups showed no significant differences. FCR was the lowest in the 1.0 and 2.0 g/kg CCPE groups but was not significantly different in the 0.5 g/kg group (*p* < 0.05). Over the entire period, ADG in the antibiotic growth promoter (AGP) group was the lowest but was not significantly different from the control group. ADFI was the lowest in the 1.0 and 2.0 g/kg CCPE groups, comparable to the AGP group. The FCR in the 0.5, 1.0, and 2.0 g/kg CCPE groups was significantly lower than that in the other groups (*p* < 0.05). From the partial budget analysis, in the groups supplemented with 1.0 and 2.0 g/kg of CCPE in the diet, feed costs were significantly lower compared to the control group (*p* < 0.05). Similarly, feed costs were reduced in the groups receiving 0.5, 1.0, and 2.0 g/kg of CCPE (*p* < 0.05). This reduction was consistent with increases in revenue, net profit and the benefit/cost ratio (*p* < 0.05).

### 3.2. Serum Biochemistry and Lipid Profiles

As presented in Table 5, the ALT levels were significantly lower in the group supplemented with 0.5 g/kg of CCPE (*p* < 0.05), with no significant differences observed in the groups supplemented with 1.0 and 2.0 g/kg of CCPE. Similarly, the AST levels were significantly lower in the 1.0 g/kg CCPE group, with no significant difference observed in the 1.0 and 2.0 g/kg CCPE groups (*p* < 0.05). Additionally, the group supplemented with 2.0 g/kg CCPE exhibited significantly lower triglyceride levels compared with the other groups (*p* < 0.05). Supplementation with CCPE at 0.5, 1.0, and 2.0 g/kg resulted in a significant increase in HDL concentrations in the serum of broiler chicks (*p* < 0.05), as depicted in Figure 1. However, there were no significant differences among the experimental groups regarding total cholesterol, LDL, globulin, creatinine, total protein, BUN, and albumin concentrations.

### 3.3. Carcass Characteristics

As presented in Table 6, the results revealed no significant differences among the experimental groups in terms of carcass percentage (%), carcass composition (%), meat percentage (%), and internal organ percentage (%). However, the groups supplemented with CCPE at 0.5, 1.0, and 2.0 g/kg diet exhibited significantly higher carcass weights than the other groups (*p* < 0.05).

### 3.4. Meat Quality

As presented in Table 7, the pH values (at 15 min, 24 h, and 48 h), meat color values (at 24 h and 48 h), cooking loss, and shear force did not differ significantly among the groups (*p* > 0.05). However, the drip loss value at 24 h was significantly lower in the groups supplemented with 0.5, 1.0, and 2.0 g/kg of CCPE compared with the other groups (*p* < 0.05). Additionally, at 48 h post-slaughter, the drip loss value in the groups supplemented with 1.0 and 2.0 g/kg of CCPE was lower than in the control group and the group supplemented with 0.25 g/kg of AGPs, but not significantly different from the group receiving 0.5 g/kg of CCPE (*p* < 0.05). In addition, the meat color values at 24 h and 48 h post-slaughter, the cooking loss, and shear force of broiler thighs did not differ significantly among the groups (*p* > 0.05).

### 3.5. Intestinal Morphology

As shown in Table 8 and Figure 2, CCPE affected the histomorphology of the duodenum, jejunum, and ileum in broiler chickens. In the duodenum, the 0.5 g/kg CCPE group had the highest VH and VH:CD ratio (*p* < 0.05). In the jejunum, the 0.5 g/kg CCPE group had the highest VH and VH:CD ratio, but was not significantly different in the 1.0 and 2.0 g/kg group (*p* < 0.05). In the ileum, the VH and VH:CD ratio were the highest in the 0.5 1.0 and 2.0 g/kg CCPE groups, but not significantly different in the control group (*p* < 0.05). No significant differences were observed in the VW and CD in the duodenum or the jejunum and ileum among the groups (*p* > 0.05).

### 3.6. Ceacal Microbiota Activity

As shown in Figure 3, broiler chicks supplemented with CCPE at 0.5, 1.0, and 2.0 g/kg diets showed significant reductions (*p* < 0.05) in both *E. coli* and *Salmonella* concentrations in the cecum compared to the control group. The effect of 2.0 g/kg CCPE was comparable to that of the antibiotic supplement in reducing *E. coli* levels (*p* < 0.05). Additionally, the cecum of chicks supplemented with CCPE at 1.0 and 2.0 g/kg had significantly higher (*p* < 0.05) concentrations of *Lactobacillus* spp. compared to both the control and antibiotic groups.

### 3.7. Immune Response and Antioxidant-Related Gene Expression

The relative expression of genes involved in immunity and antioxidant activity in the liver and intestine (jejunum) of broilers is shown in Figure 4 and Figure 5, respectively. The findings indicated that the expression of immune-related genes (proinflammatory cytokines: IL-1β, IL-6, and TNF-α) in the groups that received CCPE supplementation at 0.5, 1.0, and 2.0 g/kg exhibited a substantial decrease compared to the group that received the basal diet (*p* < 0.05). On the other hand, the groups that received CCPE supplementation at levels of 0.5, 1.0, and 2.0 g/kg showed a significant increase in the expression of antioxidant-related genes (MnSOD, CAT, GSH-Px1) compared to the group that received the basal diet (*p* < 0.05).

## 4. Discussion

The incorporation of industrial and agricultural by-products as feed additives enhances the nutritional value of low-quality materials, thereby offering valuable feed supplements. This practice not only mitigates environmental impacts but also provides economic benefits to farmers [31]. This study’s primary objective was to evaluate CCPE’s impact on broiler chickens’ production efficiency. CCPE is recognized for its antioxidant properties due to the presence of flavonoids and phenolic compounds, which are biologically active substances [3]. Additionally, CCPE exhibits antiviral, antifungal, and antimicrobial properties, enhancing the surface area available for nutrient absorption in animals. This attribute facilitates efficient food assimilation, leading to improved growth performance and increased body weight in the animals [32].

No irregularities were observed in the diets of broilers supplemented with CCPE throughout the 35-day experimental period. The findings indicate that various performance metrics, including final weight, ADG, ADFI, and FCR, were significantly improved in the CCPE supplementation groups at levels of 0.5, 1.0, and 2.0 g/kg. Specifically, the final weights, ADG, and ADFI were notably higher (*p* < 0.05) with CCPE supplementation. Moreover, the 35-day average body weight of broilers in the antibiotic control group was no better than the control group, and similar results were observed by Yang et al. [33] and Srinual et al. [22]. Additionally, a reduction in FCR in the CCPE-supplemented groups suggests a potential enhancement of feed efficiency. According to Geremu et al. [34], the bioactive compounds in coffee pulp by-products provide additional health benefits, such as polyphenols, which possess significant antioxidant and antibacterial properties. These compounds enhance the structural integrity of intestinal villi, facilitate nutrient absorption, and inhibit the growth of pathogenic bacteria in the gastrointestinal tract. Numerous studies have elucidated the role of phenolic compounds, which are the principal constituents of CCP, in enhancing chicken growth efficiency. These phenolic compounds are recognized for their potent antioxidant properties. Dietary supplementation with CGA has been documented to positively influence animal growth performance. For instance, Zhang et al. [14] demonstrated that in a 28-day experimental period, supplementation with CGA at concentrations ranging from 0.25 to 1.0 g/kg resulted in a linear improvement in both body weight and ADG in piglets. Similarly, Zha et al. [13] observed that the incorporation of CGA at 0.5 and 1.0 g/kg did not affect FCR but resulted in a linear increase in body weight, ADG, and ADFI. The observed enhancement of growth performance among poultry administered with CCPE can be partially ascribed to the concomitant improvement in antioxidant capacity. Additionally, the caffeine present in CCP exerts physiological effects on animals. Caffeine acts as an antagonist to A1 adenosine receptors in the hypothalamus, leading to appetite suppression and augmented energy use. However, high caffeine concentrations (28 mg/kg/day) have been reported to reduce plasma triiodothyronine levels, potentially impacting broiler chickens’ growth trajectories [15,35]. This study found that CCPE supplementation led to reduced total feed costs, increased revenue, and an improved net profit and benefit/cost ratio. The increased weight gain in the experimental groups, compared to the control, offset the reduced feed costs, thereby enhancing economic efficiency. Additionally, using coffee by-products helped reduce the cost associated with antibiotics as growth promoter.

The current study highlights the beneficial effects of CCPE supplementation on hematological parameters, specifically noting a reduction in TG levels and an increase in HDL. Bhandarkar et al. [36] demonstrated that chlorogenic acid, when administered at a concentration of 0.005 g/kg for 45 days, led to decreased levels of blood lipids, including TC, TG, and LDL, while enhancing HDL levels. Additionally, Kamely et al. [35] observed that caffeine levels influence TG concentrations, with a dosage of 0.05 g/kg resulting in a reduction in TG levels in broilers. Trigonelline, recognized for its antioxidant properties, mitigates endoplasmic reticulum-related stress and alleviates oxidative stress-induced damage in pancreatic cells and adipocytes. However, this study did not reveal significant variations in TC and LDL levels. Liver enzymes such as ALT, AST, and ALP are crucial indicators of hepatic function [37]. The study indicated that the average ALT and AST levels in the CCPE group were more favorable compared to the control group, suggesting a potential hepatoprotective effect of CCPE. Elevated serum hepatic enzyme levels, often a consequence of hepatocyte membrane disruption, release intracellular contents and lead to a pronounced increase in liver enzyme concentrations. This increase may be attributed to hepatocellular damage associated with the detoxification process of bacterial and pathogen toxins [37,38].

The addition of CCPE at various concentrations did not significantly affect the overall quality of meat, while carcass weights increased significantly (*p* < 0.05) by dietary CCPE 1.0 and 2.0 g/kg supplementation. Lipiński et al. [39] reported that feed supplementation with polyphenols did not influence the carcass in broilers. On the other hand, Qaid et al. [40] reported that *Rumex nervosus* has phytochemical products, such as gallic acid (GA), which improves carcass weight in broilers. However, CCPE supplementation notably reduced drip loss in breast meat at both 24 and 48 h. Lipid oxidation, a process leading to rancid odors, off-flavors, discoloration, nutritional degradation, and reduced shelf life, also poses potential health risks to consumers [41]. The observed reduction in lipid oxidation with CCPE supplementation suggests enhancement of the antioxidant properties of breast meat. This finding supports the hypothesis that bioactive compounds within CCP contribute significantly to improving the oxidative stability of meat products. Antioxidants are widely utilized to extend the shelf life of food products [42]. Bergamaschi et al. [43] reported the total phenolic content, chlorogenic acid, and caffeine levels in coffee extracts, all of which exhibit antioxidant capacities. Additionally, CCP has demonstrated significant antioxidant activity and free radical inhibition [34]. Lipid peroxidation in meat results in decreased sensitivity to hydrolysis, impaired protein degradation, reduced water retention within myofibrils, and disruption of membrane integrity in muscle cells, ultimately leading to the loss of meat juice. A reduction in cooking loss, which indicates improved water retention, appears to correlate with an increase in muscle antioxidant capacity [44].

Analysis of the intestinal microbiota revealed that supplementation with CCPE significantly reduced the abundance of pathogenic microorganisms within the gastrointestinal tract. Ashour et al. [26] investigated the effects of green coffee powder supplementation in broiler chickens and observed that coffee constituents influence gut microbial communities. Specifically, the incidence of *E. coli* and *Salmonella* decreased with CCPE supplementation. Furthermore, increasing the concentration of CCPE in the feed reduced overall microbial load compared to the control group. Notably, CCPE supplementation at levels of 1.0 and 2.0 g/kg led to an increase in the population of *Lactobacillus* spp. The antimicrobial efficacy of CCP can be attributed to several active components, including chlorogenic acid, polysaccharides, phenolic polymers, and caffeine. These constituents exhibit their bactericidal effects through multiple mechanisms, primarily by altering the membrane potential of bacterial cells and disrupting intracellular ATP homeostasis. The outer membrane of Gram-negative bacteria appears to be particularly susceptible to these effects, potentially rendering the cellular environment inhospitable for microbial proliferation.

Caffeine disrupts the lipid bilayer of microbial cell membranes, leading to cell death in *E. coli*. Its antimicrobial action primarily involves altering membrane integrity, which increases permeability and causes intracellular leakage. Ibrahim et al. [45] found that caffeine effectively inhibits *E. coli* O157, with a 0.75% concentration reducing the bacterial population by 1.4 log CFU/mL and a 1.5% concentration achieving a reduction of over 3 log CFU/mL. Chlorogenic acid exerts its antimicrobial effects by enhancing the permeability of bacterial outer and plasma membranes, leading to cell death. It has shown activity against various bacterial species, including *E. coli* and *Staphylococcus aureus*. Kabir et al. [46] demonstrated that chlorogenic acid, along with ferulic, isoferulic, benzoic, and hydroxybenzoic acids, specifically inhibited *E. coli* IFO 3301.

Intestinal morphology is a crucial indicator of gastrointestinal health, encompassing factors such as the integrity of the intestinal barrier, the efficacy of nutrient digestion, and the absorptive capacity of the small intestine [47,48]. Supplementation with CGA has been shown to positively influence various morphological parameters of the intestine, including villus height, crypt depth, and the ratio of villus height to crypt depth. Liu et al. [12] observed that administering CGA at a concentration of 0.125 g/kg improved the VH ratio in both the duodenum and jejunum. Liu et al. [48] demonstrated that improvements in intestinal morphology, facilitated by CGA supplementation, led to enhanced growth performance. Additionally, CGA supplementation resulted in an increased villus height and VH:CD ratio, coupled with a reduction in ileal crypt depth in broilers subjected to oxidative stress. Liu et al. [12] observed that administering CGA at a concentration of 0.125 g/kg improved the VH ratio in both the duodenum and the jejunum. Similarly, Liu et al. [11] reported that a dosage of 1.0 g/kg CGA, when administered in conjunction with *Eimeria* infection, resulted in villus heights in the jejunum and duodenum comparable to those observed in control groups fed a basal diet. Furthermore, Qosimah et al. [49] found that coffee extract administered at doses of 500 and 1000 mg/kg significantly enhanced intestinal structural integrity and increased villus length. This effect was particularly helpful in alleviating damage caused by *Salmonella enteritidis* bacterial infection. In addition, Liu et al. [11] identified diamine oxidase [41] and D-lactic acid levels in the bloodstream as reliable biomarkers for assessing intestinal permeability and barrier integrity. Supplementation with CGA has been shown to reduce diamine oxidase (DAO) levels, suggesting that CGA may improve growth and mitigate intestinal barrier damage by enhancing intestinal permeability and morphology in broiler chickens. The presence of CGA in coffee pulp appears to be adequate to effectuate significant improvements in intestinal morphology.

Regarding immune-related gene expression, dietary CCP supplementation resulted in notable downregulation of genes associated with proinflammatory cytokines, specifically IL-1β, IL-6, and TNF-α. Concurrently, there was a significant upregulation of genes related to antioxidant defense, including manganese superoxide dismutase (MnSOD), catalase [50], and glutathione peroxidase 1 (GSH-Px1). Activated macrophages are the principal source of proinflammatory cytokines, which are pivotal in the exacerbation of inflammatory responses. Extensive evidence underscores the involvement of inflammatory cytokines such as TNF-α, IL-1β, and IL-6 in the pathogenesis of pathological pain [51,52]. CCP is rich in flavonoids and phenolic compounds, which are biologically active constituents with antioxidant properties. These antioxidants can modulate the activity of transcription factors involved in the immune response. They can also reduce the synthesis of proinflammatory cytokines and inhibit critical signaling pathways and enzymes integral to immunological processes [53,54]. Consequently, there is a reduction in the expression of cytokines that typically facilitate inflammation.

## 5. Conclusions

This study demonstrated that CCPE enhanced broiler chickens’ growth performance. Supplementation with CCPE also reduced serum triglycerides, ALT, and AST levels. CCPE positively influenced gut microbiota by decreasing pathogenic microorganisms and improving small intestinal morphology. Additionally, CCPE reduced drip loss in breast meat and enhanced the expression of antioxidant-related genes. Based on these findings, it is recommended that broiler diets include CCPE broiler diets should include CCPE at a level of 1.0 g/kg diet.

## Figures and Tables

**Figure 1 animals-15-00244-f001:**
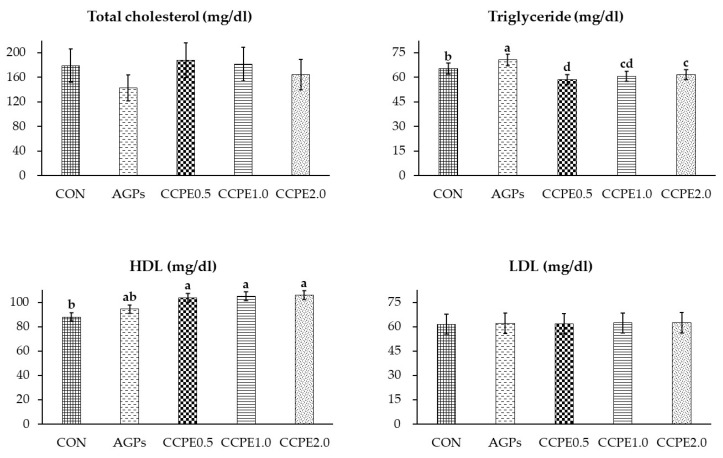
Effects of coffee cherry pulp extract on lipid profile. CON: control group with basal diet; AGPs: antibiotic growth promoter (AGP) group; CCPE 0.5: coffee cherry pulp extract at 0.5 g/kg diet; CCPE 1.0: coffee cherry pulp extract at 1.0 g/kg diet; CCPE 2.0: coffee cherry pulp extract at 2.0 g/kg diet; HDL: high-density lipoprotein; LDL: low-density lipoprotein. ^a, b, c, d^ Means with different superscripts are significantly different at *p* < 0.05.

**Figure 2 animals-15-00244-f002:**
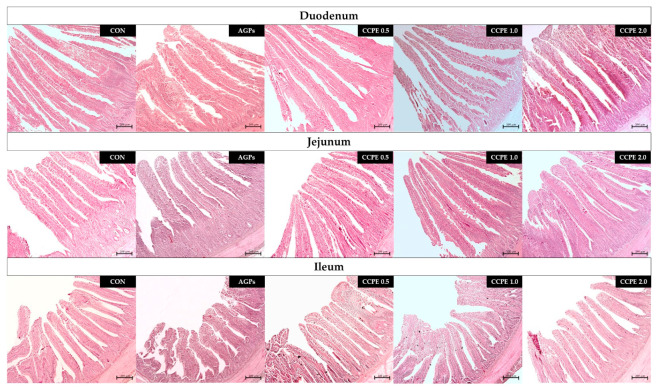
Histological representations of the H&E-stained duodenum, jejunum, and ileum sections of broiler chickens. CON: control group with basal diet; AGP: antibiotic growth promoter (AGP) group; CCPE 0.5: coffee cherry pulp extract at 0.5 g/kg diet; CCPE 1.0: coffee cherry pulp extract at 1.0 g/kg diet; CCPE 2.0: coffee cherry pulp extract at 2.0 g/kg diet. Magnification was 10× the objective lens. Scale bars represent 200 µm.

**Figure 3 animals-15-00244-f003:**
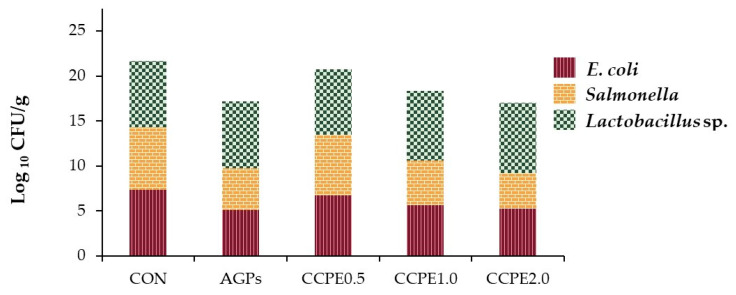
Cecum microbial count of broiler chickens impacted by dietary supplements with different levels of coffee cherry pulp extract. CON: control group with basal diet; AGP: antibiotic growth promoter group; CCPE 0.5: coffee cherry pulp extract at 0.5 g/kg diet; CCPE 1.0: coffee cherry pulp extract at 1.0 g/kg diet; CCPE 2.0: coffee cherry pulp extract at 2.0 g/kg diet.

**Figure 4 animals-15-00244-f004:**
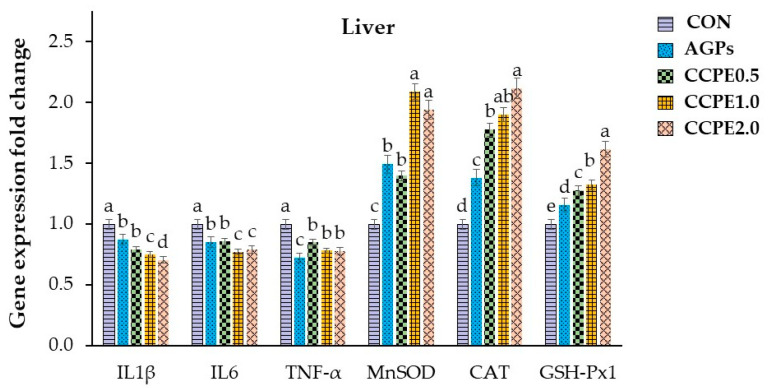
Expressions of antioxidant- and immune-related genes in the liver of broilers fed with coffee cherry pulp extract. Three replicates. IL-1β: interleukin 1 beta; IL6: interleukin 6; TNF-α: tumor necrosis factor alpha; MnSOD: manganese-containing superoxide dismutase; CAT: catalase; GSH-Px1: glutathione peroxidase 1. ^a, b, c, d, e^ Means with different superscripts are significantly different at *p* < 0.05.

**Figure 5 animals-15-00244-f005:**
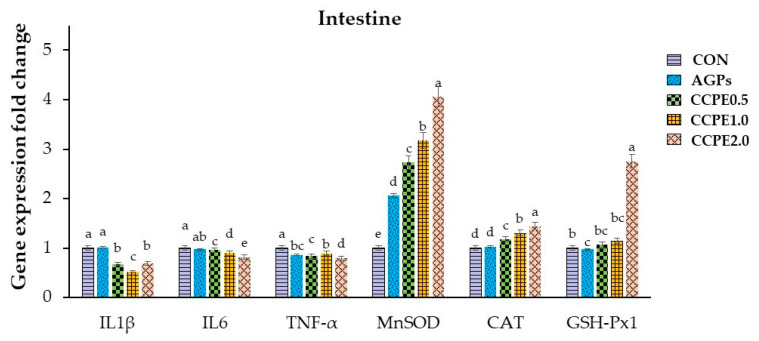
Expressions of antioxidant- and immune-related genes in the ileum of broilers fed with coffee cherry pulp extract. Three replicates. IL-1β: interleukin 1 beta; IL6: interleukin 6; TNF-α: tumor necrosis factor alpha; MnSOD: manganese-containing superoxide dismutase; CAT: catalase; GSH-Px1: glutathione peroxidase 1. ^a, b, c, d, e^ Means with different superscripts are significantly different at *p* < 0.05.

**Table 1 animals-15-00244-t001:** Total phenolic compounds (mg/g), chlorogenic acid (mg/g), gallic acid (mg/g), and caffeine (mg/g) content of coffee cherry pulp extract.

	CCPE
Total phenolic compounds	378.33 ± 17.62
Chlorogenic acid	1015.61 ± 1.69
Gallic acid	91.22 ± 6.06
Caffeine	254.65 ± 2.09

The average of three replicate experiments ± SD is used to determine the results.

**Table 2 animals-15-00244-t002:** Ingredients and nutrient composition of broiler starter and finisher diets.

	Starter (1–14 d)	Finisher (15–35 d)
Ingredients (%)		
Yellow corn	57.17	59.58
Raw rice bran	5.00	5.00
Soybean meal (48% CP)	28.62	20.74
Rapeseed meal (38% CP)	3.20	6.00
Pork meal (50% CP)	3.00	4.00
Soybean oil	-	2.94
Salt	0.26	0.14
L-lysine	0.10	0.08
DL-methionine	0.25	0.15
Biofoss (21% P) ^1^	0.70	0.02
Calcium carbonate	1.07	0.64
Premixes ^2^	0.63	0.71
Cornstarch	0.10	0.10
Calculated composition		
Metabolizable energy (kcal/kg)	2987.89	3095.98
Calcium (%)	0.61	0.41
Available phosphorus (%)	0.59	0.39
Sodium (%)	0.20	0.18
Chlorine (%)	0.27	0.23
Digestible lysine (%)	1.20	1.12
Digestible Met (%)	0.57	0.48
Digestible Met + Cys (%)	0.92	0.80
Digestible tryptophan (%)	0.26	0.25
Digestible linoleic (%)	1.56	1.64
Choline (mg/kg)	956.90	927.80

^1^ Biofoss supplies per kg of diet: monocalcium phosphate 1000 g, calcium 150 g, phosphorus 210 g and fluorine 0.21%. ^2^ Mineral premix supplies per kg of diet: vitamin A 20,000,000 IU, vitamin D3 4,000,000 IU, vitamin E 11,000 IU, vitamin K3 4.00 g, vitamin B1 5.00 g, vitamin B2 10.00 g, vitamin B6 5.00 g, vitamin B12 0.02 g, vitamin C 15.00 g, pantothenic acid 15.00 g, folic acid 3.00 g, nicotinic acid 40.00 g, biotin 0.20 g, magnesium 100.00 g, potassium 90.00 g, sodium 100.00 g, and feed additive 25.30 g.

**Table 3 animals-15-00244-t003:** Oligonucleotide sequences of sense and antisense primers for real-time PCR products used in this study.

Target Gene	Primer Sequences	Accession Number	Product Size (bp)
β-actin	F: GAGAAATTGTGCGTGACATCA	L08165	152
	R: CCTGAACCTCTCATTGCCA		
IL-1β	F: TGGGCATCAAGGGCTACA	Y15006	244
	R: TCGGGTTGGTTGGTGATG		
IL-6	F: CAAGGTGACGGAGGAGGAC	AJ309540	254
	R: TGGCGAGGAGGGATTTCT		
TNF-α	F: TGTGTATGTGCAGCAACCCGTAGT	NM204267	229
	R: GGCATTGCAATTTGGACAGAAGT		
MnSOD	F: CACTCTTCCTGACCTGCCTTACG	NM204211	146
	R: TTGCCAGCGCCTCTTTGTATT		
CAT	F: CTGTTGCTGGAGAATCTGGGTC	NM001031215	160
	R: TGGCTATGGATGAAGGATGGAA		
GSH-Px1	F: GCGACTTCCTGCAGCTCAACGA	GQ502186.2	99
	R: CGTTCTCCTGGTGCCCGAAT		

IL: interleukin; TNF-α: tumor necrosis factor alpha; MnSOD: manganese-containing superoxide dismutase; CAT: catalase; GSH-Px1: glutathione peroxidase 1.

**Table 4 animals-15-00244-t004:** Growth performance parameters of broiler chickens of different ages as influenced by dietary supplementation with different coffee cherry pulp extract levels.

Parameter	CON	AGPs, g/kg	Coffee Cherry Pulp Extract, g/kg			SEM	*p*-Value
		0.25	0.5	1.0	2.0		
1–14 days							
Initial weight (g)	46.40	46.40	46.20	46.80	46.40	0.337	0.990
Final weight (g)	551.20 ^b^	536.40 ^b^	650.50 ^a^	658.60 ^a^	653.20 ^a^	13.223	0.001
ADG (g/d)	36.86 ^b^	35.00 ^b^	43.16 ^a^	43.70 ^a^	43.34 ^a^	1.040	0.003
ADFI (g/d)	33.82	34.37	32.21	31.68	34.67	0.429	0.088
FCR	0.93 ^ab^	0.99 ^a^	0.75 ^c^	0.73 ^c^	0.81 ^bc^	0.027	0.001
15–35 days							
Initial weight (g)	551.20 ^b^	536.40 ^b^	650.50 ^a^	658.60 ^a^	653.20 ^a^	13.223	0.001
Final weight (g)	1760.00 ^b^	1720.00 ^b^	1980.00 ^a^	2000.00 ^a^	2000.00 ^a^	32.619	0.001
ADG (g/d)	57.56 ^bc^	55.41 ^c^	66.37 ^a^	63.93 ^ab^	63.20 ^ab^	1.255	0.014
ADFI (g/d)	121.80 ^a^	113.20 ^ab^	121.87 ^a^	108.11 ^b^	105.31 ^b^	2.178	0.026
FCR	2.12 ^a^	2.06 ^ab^	1.84 ^a^	1.71 ^b^	1.68 ^b^	0.055	0.018
1–35 days							
Initial weight (g)	46.40	46.40	46.20	46.80	46.40	0.337	0.990
Final weight (g)	1760.00 ^b^	1720.00 ^b^	1980.00 ^a^	2000.00 ^a^	2000.00 ^a^	32.619	0.001
ADG (g/d)	48.96 ^bc^	46.67 ^c^	57.09 ^a^	54.70 ^ab^	55.82 ^ab^	1.234	0.013
ADFI (g/d)	86.60 ^a^	81.67 ^ab^	86.00 ^a^	77.54 ^b^	77.06 ^b^	1.187	0.008
FCR	1.77 ^a^	1.77 ^a^	1.51 ^b^	1.42 ^b^	1.40 ^b^	0.045	0.001
Economic evaluation							
Total feed cost/chick, USD	1.73 ^a^	1.63 ^ab^	1.71 ^a^	1.55 ^b^	1.54 ^b^	0.024	0.008
Feed cost/kg BW gain, USD	1.01 ^a^	1.01 ^a^	0.86 ^b^	0.81 ^b^	0.79 ^b^	0.025	0.001
Total revenue, USD	2.01 ^b^	1.96 ^b^	2.26 ^a^	2.28 ^a^	2.28 ^a^	0.037	0.001
Net profit, USD	0.28 ^c^	0.33 ^c^	0.54 ^b^	0.73 ^a^	0.74 ^a^	0.047	0.001
Benefit/cost ratio	1.17 ^c^	1.20 ^bc^	1.31 ^b^	1.48 ^a^	1.48 ^a^	0.031	0.001

ADG: average daily gain; ADFI: average daily feed intake; CON: basal diet control; FCR: feed conversion ratio; BW: body weight; SEM: standard error of the mean. ^a, b, c^ Means within a row with different superscripts are significantly different at *p* < 0.05.

**Table 5 animals-15-00244-t005:** The effect of coffee cherry pulp extract supplementation on serum biochemistry of broiler chicks at three levels (0.5, 1.0, and 2.0 g/kg diet) compared to the control.

Parameter	CON	AGPs, g/kg	Coffee Cherry Pulp Extract, g/kg			SEM	*p*-Value
		0.25	0.5	1.0	2.0		
ALT (U/L)	3.67 ^a^	3.00 ^ab^	2.00 ^c^	2.33 ^bc^	2.67 ^bc^	0.182	0.009
Globulin (g/dl)	2.13	1.97	1.97	1.90	2.20	0.048	0.246
ALP (U/L)	1864.00	1771.33	2082.67	2083.33	2020.33	68.167	0.554
Creatinine (mg/dl)	0.29	0.52	0.49	0.31	0.37	0.035	0.116
Total protein (g/dl)	3.33	3.27	3.23	3.00	3.60	0.070	0.071
BUN (mg/dl)	1.20	0.73	1.00	1.30	0.87	0.101	0.413
Albumin (g/dl)	1.20	1.30	1.27	1.10	1.40	0.036	0.074
AST (U/L)	337.00 ^ab^	351.67 ^a^	284.00 ^bc^	244.67 ^c^	291.33 ^bc^	12.150	0.008

ALT: alanine transaminase; ALP: alkaline phosphatase; BUN: blood urea nitrogen; AST: aspartate aminotransferase; SEM: standard error of mean. ^a, b, c^ Means within a row with different superscripts are significantly different at *p* < 0.05.

**Table 6 animals-15-00244-t006:** Carcass characteristics of broiler chicks as affected by graded levels of dietary coffee cherry pulp extract supplementation.

Parameter	CON	AGPs, g/kg	Coffee Cherry Pulp Extract, g/kg			SEM	*p*-Value
		0.25	0.5	1.0	2.0		
Live weight (g)	1760.00 ^b^	1720.00 ^b^	1980.00 ^a^	2000.00 ^a^	2000.00 ^a^	32.619	0.001
Defeathered weight (g)	1640.00 ^b^	1640.00 ^b^	1860.00 ^a^	1920.00 ^a^	1920.00 ^a^	33.902	0.001
Carcass weight (g)	1471.14 ^b^	1473.10 ^b^	1678.92 ^a^	1738.06 ^a^	1747.62 ^a^	32.855	0.001
Carcass percentage (%)	84.49	85.58	84.74	86.95	87.42	0.469	0.180
Carcass composition (% lw)							
Neck	4.83	5.22	6.82	4.48	4.64	0.633	0.804
Head	3.27	3.29	3.30	3.24	3.29	0.056	0.998
Wing	8.24	7.94	7.62	7.58	7.56	0.108	0.197
Drumstick	10.35	10.41	10.31	10.27	10.34	0.066	0.976
Shank	3.53	3.38	3.42	3.38	3.58	0.045	0.527
Skeleton	19.39	19.69	19.74	20.17	18.77	0.293	0.682
Meat percentage (% lw)							
Breast	12.55	12.37	13.18	13.49	13.24	0.190	0.284
Thigh	13.48	12.90	13.65	13.73	14.03	0.144	0.141
Internal organ percentage (% lw)							
Liver	2.06	2.41	2.15	1.95	2.02	0.059	0.108
Spleen	0.09	0.09	0.08	0.07	0.07	0.004	0.373
Heart	0.51	0.41	0.46	0.42	0.41	0.016	0.169
Proventriculus	0.42	0.38	0.37	0.36	0.42	0.016	0.700
Gizzard	3.08	2.88	2.59	2.71	2.47	0.076	0.080
Small intestine	3.52	3.59	3.69	3.53	3.24	0.064	0.242
Cecum	0.76	0.73	0.59	0.50	0.54	0.035	0.062
Abdominal fat	0.53	0.31	0.37	0.43	0.44	0.050	0.732

SEM: standard error of mean. ^a, b^ Means within a row with different superscripts are significantly different at *p* < 0.05.

**Table 7 animals-15-00244-t007:** Effect of coffee cherry pulp extract on breast and thigh meat quality in broiler chickens.

Parameter	CON	AGPs, g/kg	Coffee Cherry Pulp Extract, g/kg			SEM	*p*-Value
		0.25	0.5	1.0	2.0		
pH 15 min	6.02	5.87	5.47	5.54	5.81	0.122	0.62
pH 24 h	6.29	6.38	6.24	6.49	6.08	0.191	0.976
pH 48 h	5.55	5.39	5.37	5.4	5.48	0.039	0.561
Breast color (24 h)							
Lightness (L*)	54.07	54.73	53.98	52.43	54.35	0.384	0.405
Redness (a*)	0.16	0.26	0.2	0.63	0.11	0.179	0.917
Yellowness (b*)	12.05	11.68	11.12	11.82	11.44	0.28	0.88
Thigh color (24 h)							
Lightness (L*)	54.2	55.75	54.38	54.67	55.42	0.242	0.075
Redness (a*)	1.36	0.15	0.34	0.85	0.64	0.17	0.188
Yellowness (b*)	11.58	10.85	9.33	9.88	10.65	0.501	0.692
Breast color (48 h)							
Lightness (L*)	53.59	54.3	53.69	53.2	51.08	0.751	0.741
Redness (a*)	0.43	0.74	0.47	1.25	0.1	0.158	0.196
Yellowness (b*)	12.97	13.59	12.36	12.23	10.84	0.389	0.24
Thigh color (48 h)							
Lightness (L*)	54.1	55.53	55.32	54.69	55.1	0.201	0.222
Redness (a*)	1.4	0.38	0.28	1.77	0.22	0.239	0.109
Yellowness (b*)	13.08	11.52	10.02	11.38	11.69	0.359	0.103
Water holding capacity							
Breast							
Drip loss (%, 24 h)	1.86 ^a^	2.08 ^a^	1.24 ^b^	1.27 ^b^	1.21 ^b^	0.091	<0.001
Drip loss (%, 48 h)	3.40 ^a^	3.49 ^a^	2.85 ^ab^	2.16 ^b^	2.27 ^b^	0.161	0.004
Shear force (kgf/cm^2^)	3.28	3.28	3.31	3.29	3.28	0.007	0.392
Cooking loss (%)	25.43	25.36	25.4	25.43	25.33	0.023	0.577
Thigh							
Shear force (kgf/cm^2^)	4.08	4.08	4.08	4.11	4.09	0.007	0.392
Cooking loss (%)	20.03	19.96	20.00	20.03	19.93	0.023	0.577

SEM: standard error of mean. ^a, b^ Means within a row with different superscripts are significantly different at *p* < 0.05.

**Table 8 animals-15-00244-t008:** Impact of dietary supplementation with coffee cherry pulp extract on intestinal morphology.

Items	CON	AGPs, g/kg	Coffee Cherry Pulp Extract, g/kg			SEM	*p*-Value
		0.25	0.5	1.0	2.0		
**Duodenum**							
VH (µm)	1068.19 ^e^	1087.90 ^d^	1163.36 ^a^	1137.18 ^b^	1113.27 ^c^	1.509	0.001
VW (µm)	190.34	187.18	191.04	192.36	195.47	1.731	0.555
CD (µm)	270.61	280.59	246.50	252.97	256.68	5.033	0.202
VH:CD	3.99 ^b^	3.93 ^b^	4.83 ^a^	4.53 ^ab^	4.46 ^ab^	0.096	0.007
**Jejunum**							
VH (µm)	1036.66 ^d^	1057.73 ^c^	1102.07 ^a^	1085.65 ^ab^	1075.32 ^ab^	3.704	0.001
VW (µm)	185.49	168.82	186.87	175.27	169.73	4.277	0.555
CD (µm)	274.68	275.32	252.93	258.25	258.00	3.427	0.084
VH:CD	3.83 ^b^	3.90 ^b^	4.36 ^a^	4.21 ^ab^	4.18 ^ab^	0.054	0.002
**Ileum**							
VH (µm)	716.57 ^ab^	675.95 ^b^	768.07 ^a^	755.56 ^a^	772.74 ^a^	9.178	0.001
VW (µm)	146.94	145.77	147.53	144.91	144.40	1.778	0.966
CD (µm)	136.59	131.01	136.26	136.22	135.65	1.098	0.552
VH:CD	5.28 ^ab^	5.17 ^b^	5.65 ^a^	5.56 ^a^	5.70 ^a^	0.074	0.039

SEM: standard error of mean. ^a, b, c, d, e^ Means within a row with different superscripts are significantly different at *p* < 0.05. VH: villus height; VW: villus width; CD: crypt depth; VH:CD: villus height per crypt depth ratio.

## Data Availability

Data are available on request due to restrictions, e.g., privacy or ethical restrictions. The data presented in this study are available on request from the corresponding author. The data are not publicly available due to the law of the Ministry of Higher Education, Science, Research, and Innovation.

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
