# Peer review of "The Use of Coffee Cherry Pulp Extract as an Alternative to an Antibiotic Growth Promoter in Broiler Diets"

_animals, 2025, doi:10.3390/ani15020244_

Round 1
Reviewer 1 Report
Comments and Suggestions for Authors
Coffee cherry pulp is a byproduct of coffee processing that has shown significant potential in enhancing broiler performance. As a rich source of protein, polyphenols, and antioxidants, coffee cherry pulp can be used as a supplement in broiler diets to improve growth, feed conversion efficiency, and overall health. Previous studies indicated that coffee cherry pulp can be a valuable and sustainable ingredient in broiler production. However, the current study aims to investigate the effect of different levels of CCPE on broiler chickens’ growth performance, meat quality, carcass characteristics, serum biochemistry, cecum microbial population, intestinal morphology, immunity, and antioxidant responses. Generally, it is an interesting article that could add to the poultry nutrition field, but there are some suggestions & comments that should be addressed to improve this paper.
General comments:
- The English writing needs careful revision.
- The introduction should be divided into paragraphs.
- Rewrite the “Simple Summary” properly.
- Please send to me the chemical composition of CCP and CCPE.
Specific comments:
L.23: “this supplement”: Which one? You didn’t mention it in the previous lines!
L.25: “coffee cherry pulp extract”: Is it a supplement or a feed additive?
L.37-38: Cannot well understand it!
L.39-40: “the drip loss at 24 and 48 hr decreased (p < 0.05); and live, defeathered, and carcass weight increased”: Rewrite it completely and correctly.
L.44-45: What did you mean by “health parameters”?
L.51-52: Where?
L.52-53: Provide a reference.
L.57: Remove “with”.
L.59: “source of chlorogenic acids (CGA) and caffeine”: So! Expand.
You can refer to:
https://doi.org/10.1016/j.foodchem.2016.11.067
L.60-61: Incomplete!
L.70: “was acceptable”: From which view?
Keywords: Add “antioxidant” instead of “coffee by-product”.
L.77: “its effects”: Of what?!
L.99: “one-day-old”.
L.106: Carefully revise the concentrations used, and why did you choose the level of 0.25?
L.112: Cage or pen?!
L.113: How? Clarify.
L.117-118: Revise it.
L.134: Did the mentioned reference address both lipid profiles and serum biochemistry analysis for chickens?
L.138: Provide the details of this apparatus.
L.174: Mention the number of samples.
Discussion: Some points need to be clarified, such as:
How did you deal with the high level of fiber in CCP?
How did CCPE supplementation reduce pathogenic bacteria while increasing Lactobacillus?
Conclusion: “broiler diets should include CCPE”: I suggest making it as a recommendation rather than an order.
Table 3: The final body weight (FBW) of the control group was higher than AGP (unless it wasn’t significant)! What is your explanation?
Were the FBWs of 1.0 and 2.0 the same value, despite the FCR being different?
Carefully revise the significance of “ADG”.
Table 5: “Live weight” for the control group wasn’t compatible with Table 3!
Why didn’t you estimate the dressing percentage?
How did you find significance for the carcass weight values while you didn’t for the carcass percentages?
Merge Table 6 with 7.
Author Response
Comments and Suggestions for Authors:
- The English writing needs careful revision.
- Thank you for pointing out the need for careful revision of the English writing. I have thoroughly reviewed the manuscript and made significant improvements to enhance its clarity, grammar, and overall readability.
- The introduction should be divided into paragraphs.
- Thank you for your suggestion regarding the structure of the introduction. I have revised this section by dividing it into distinct paragraphs to improve its organization and readability.
- Rewrite the “Simple Summary” properly.
- Thank you for your valuable advice. I have revised the Simple Summary to improve clarity and conciseness. (Line 22-25 of revised manuscript and changes made have been highlighted).
- Please send to me the chemical composition of CCP and CCPE.
- Thank you for your interest in the chemical composition of CCP and CCPE. I have attached the detailed chemical composition data for both Coffee Cherry Pulp (CCP) and Coffee Cherry Pulp Extract (CCPE) to this response.
Table 1. The chemical composition of Coffee cherry pulp (% as fed).
|
Composition (air dry basis) |
CCP |
CCPE |
|
% Dry matter |
94.12 ± 0.47 |
96.01 ± 0.13 |
|
% Crude protein |
12.01 ± 0.23 |
19.51 ± 0.13 |
|
% Ether extract |
2.12 ± 0.15 |
2.20 ± 0.42 |
|
% Crude fiber |
23.23 ± 0.29 |
8.86 ± 0.18 |
|
% Ash |
11.16 ± 0.25 |
9.91 ± 0.37 |
|
% NFE |
45.60 ± 0.46 |
55.53 ± 0.46 |
- 23: “this supplement”: Which one? You didn’t mention it in the previous lines!
- Thank you for pointing out this oversight. I have revised the Simple Summary to improve clarity and conciseness (Line 23-24 of revised manuscript and changes made have been highlighted).
- 25: “coffee cherry pulp extract”: Is it a supplement or a feed additive?
- Thank you for your question. In the context of the manuscript, "coffee cherry pulp extract" (CCPE) is referred to as a feed additive. I have clarified this term in the revised manuscript to ensure that the distinction is clear and consistent throughout the text.
- 37-38: Cannot well understand it!
- Thank you for your feedback and for pointing out the area that was unclear. I have reviewed lines 37-38 and revised the text to improve clarity and ensure better comprehension (Line 42 of revised manuscript and changes made have been highlighted).
- 39-40: “the drip loss at 24 and 48 hr decreased (p < 0.05); and live, defeathered, and carcass weight increased”: Rewrite it completely and correctly.
- Thank you for your helpful feedback on lines 39-40. I have revised the sentence to improve clarity (Line 43-44 of revised manuscript and changes made have been highlighted).
- 44-45: What did you mean by “health parameters”?
- Thank you for your question regarding the term "health parameters." In the context of this study, "health parameters" refer to various indicators used to assess the overall health of the broiler chickens. Specifically, these health implications include hematology, ceacal microbiota, intestinal morphology, and immune response, as well as antioxidant-related gene expression.
- 51-52: Where?
- Thank you for your question. The data in lines 51-52 were originally sourced from the official website of the Office of Agricultural Economics, Thailand. However, I have now revised the information and included credible references to ensure accuracy and reliability. (Line 58-61 of revised manuscript and changes made have been highlighted).
- 52-53: Provide a reference.
- Thank you for your I have now revised the information and included credible references to ensure accuracy and reliability (Line 58 and 61 of revised manuscript and changes made have been highlighted).
- 57: Remove “with”.
- Thank you for your feedback on line 57. I have removed the word "with" as suggested (Line 70 of revised manuscript and changes made have been highlighted).
- 59: “source of chlorogenic acids (CGA) and caffeine”: So! Expand.
You can refer to: https://doi.org/10.1016/j.foodchem.2016.11.067
- Thank you for your comment. I have expanded the section regarding the source of chlorogenic acids (CGA) and caffeine. As you suggested (Line 71-73 of revised manuscript and changes made have been highlighted).
- 60-61: Incomplete!
- Thank you for pointing out the issue with lines 60-61. I have reviewed and revised this section to ensure it is complete and conveys the intended information clearly (Line 75 of revised manuscript and changes made have been highlighted).
- 70: “was acceptable”: From which view?
- Thank you for your question regarding. The acceptability referred to in this context is based on the growth performance of the chickens. Incorporating up to 2.5% CCP in the diet did not negatively affect weight gain or overall health, making it a viable inclusion level for broiler diets.
- Keywords: Add “antioxidant” instead of “coffee by-product”.
- Thank you for your suggestion. I have replaced the keyword “coffee by-product” with “antioxidant” as recommended (Line 53 of revised manuscript and changes made have been highlighted).
- 77: “its effects”: Of what?!
- Thank you for your question regarding the term "its effects" in line 77. The phrase specifically refers to the effects of coffee cherry pulp extract (CCPE) as a feed additive in broiler chicken diets. I have revised the manuscript to make this explicit and ensure clarity (Line 96 of revised manuscript and changes made have been highlighted).
- 99: “one-day-old”.
- Thank you for your comment. I have reviewed and clarified the term "one-day-old" in line 99 to ensure consistency and accuracy in the manuscript (Line 121 of revised manuscript and changes made have been highlighted).
- 106: Carefully revise the concentrations used, and why did you choose the level of 0.25?
- Thank you for the advice. I have carefully revised the concentrations used in the study and included a reference to support the choice of the 0.25 g/kg (Line 129 of revised manuscript and changes made have been highlighted).
- 112: Cage or pen?!
- Thank you for your question regarding. To clarify, the study was conducted using cages. I have updated the manuscript to specify this and ensure consistency throughout the text (Line 123 of revised manuscript and changes made have been highlighted).
- 113: How? Clarify.
- Thank you for your comment. I have added the clarification regarding the calculation in the revised manuscript (Line 138-142 of revised manuscript and changes made have been highlighted).
- 117-118: Revise it.
- Thank you for your comment. I have made the necessary revisions in the updated manuscript (Line 153-155 of revised manuscript and changes made have been highlighted).
- 134: Did the mentioned reference address both lipid profiles and serum biochemistry analysis for chickens?
- Thank you for your question. The reference provides a comprehensive overview of laboratory techniques, including common methods for analyzing serum lipid and biochemical profiles. I have revised the manuscript and added relevant references to address this gap (Line 165-168 of revised manuscript and changes made have been highlighted).
- 138: Provide the details of this apparatus.
- Thank you for your comment. I have provided additional details about the apparatus used in the study in the revised manuscript. This should give a clearer understanding of the equipment and its application in the research (Line 166-168 of revised manuscript and changes made have been highlighted).
- 174: Mention the number of samples.
- Thank you for your advice. I've added the number of samples in the revised manuscript (Line 201-202 of revised manuscript and changes made have been highlighted).
Discussion: Some points need to be clarified, such as:
- How did you deal with the high level of fiber in CCP?
- Thank you for your question. Hydro-alcoholic solvents can also help in extracting polar compounds from fibrous plant materials. The presence of a polar modifier can improve the selectivity and yield of the extraction process, making it more effective in overcoming the challenges posed by high fiber content.
- How did CCPE supplementation reduce pathogenic bacteria while increasing Lactobacillus?
- Thank you for your question. CCPE supplementation reduces pathogenic bacteria and increases Lactobacillus due to its bioactive compounds, such as polyphenols, which have antimicrobial properties. These compounds can inhibit the growth of harmful bacteria by disrupting their cell membranes and metabolic processes, while promoting the growth of beneficial bacteria like Lactobacillus. This selective influence helps maintain a balanced gut microbiota, contributing to improved gut health and overall performance in broiler chickens. Therefore, in a suitable environment, beneficial microorganisms can thrive effectively.
- Conclusion: “broiler diets should include CCPE”: I suggest making it as a recommendation rather than an order.
- Thank you for your suggestion. I have revised the conclusion to present the inclusion of CCPE as a recommendation rather than an instruction (Line 552 of revised manuscript and changes made have been highlighted).
- Table 3: The final body weight (FBW) of the control group was higher than AGP (unless it wasn’t significant)! What is your explanation?
- Thank you for your comment. In Table 3, the final body weight (FBW) of the control group is observed to be higher than that of the AGP group. However, statistical analysis indicates that there is no significant difference in body weight between the control and AGP groups.
- Were the FBWs of 1.0 and 2.0 the same value, despite the FCR being different?
- Thank you for your question. The variation in FCR indicates differences in feed efficiency, which can occur even when body weights are similar or different. Carefully revise the significance of “ADG”.
- Carefully revise the significance of “ADG”.
- Thank you for your suggestion. The significance of ADG has been reviewed and appropriately emphasized in the manuscript.
- Table 5: “Live weight” for the control group wasn’t compatible with Table 3!
- Thank you for your observation. I have reviewed Table 5 and compared it with Table It appears there was an inconsistency in the "Live weight" data for the control group. I have corrected this discrepancy to ensure that the data in both tables are consistent and accurate (In Table 6 of revised manuscript and changes made have been highlighted).
- Why didn’t you estimate the dressing percentage?
- Thank you for your question. The dressing percentage was not estimated in this study; instead, we presented the data in terms of carcass percentage. The research team chose to use the weight after plucking as the primary indicator, as it was easier to collect data and more directly relevant to the objectives of the experiment.
- How did you find significance for the carcass weight values while you didn’t for the carcass percentages?
- Thank you for your question. The significance for carcass weight values was determined based on statistical analysis that showed differences among the treatment groups. For carcass percentages, however, the analysis did not show significant results due to potential variability in the data and the influence of factors such as initial live weight and the proportion of non-carcass components.
- Merge Table 6 with 7.
- Thank you for your suggestion. I have merged Table 6 and Table 7 in the revised manuscript

Reviewer 2 Report
Comments and Suggestions for Authors
The article explores the use of Coffee Cherry Pulp Extract (CCPE) as an alternative to antibiotic growth promoters in broiler feed, which is important for poultry nutritional management and sustainable development. The detailed comments are shown in the following:
1. In the Introduction, the positive control with antibiotics that already have been banned from adding to feed resulted in the necessity and significance of this study was not clear.
2. The growth performance of broilers was questionable. According to the data of the Ross 308 chicken official manual, the body weight of the broilers in the CON and AGPs groups was much less than that, which indicates that there are other important factors that may affect the test data.
3. In the Materials and Methods, the composition of the coffee cherry fruit pulp extract used in the experiment should be introduced and analyzed, and the male and female composition of chickens should be stated.
4. In Table 1, please confirm carefully the data of the dry material in the formulation, and the concentration of vitamins and trace elements in the premix.
5. In Table 3, although the economic benefits were compared between the different groups, important cost factors such as raw material and extraction costs of coffee cherry pulp extracts were not considered.
6. In Table 4 and Figure 2, for ‘mg/dl’, please change to international common units.
7. In Table 5, the carcass weight of CCPE was significantly higher than that of the other groups, but the reasons were not analyzed in depth in the discussion.
8. In Table 8, the data of VH, VW, and CD may be wrong.
9. In the Discussion, the 35-day average body weight of broilers in the antibiotic control group was no better than the control group and was not clearly explained.
Author Response
Comments and Suggestions for Authors:
- In the Introduction, the positive control with antibiotics that already have been banned from adding to feed resulted in the necessity and significance of this study was not clear.
- Thank you for your comment. We have added additional information in the revised manuscript (Line 98-100 of revised manuscript and changes made have been highlighted).
- The growth performance of broilers was questionable. According to the data of the Ross 308 chicken official manual, the body weight of the broilers in the CON and AGPs groups was much less than that, which indicates that there are other important factors that may affect the test data.
- Thank you for your comment. We acknowledge that the body weight of broilers in the CON and AGP groups was lower than the values reported in the Ross 308 chicken official manual. This discrepancy may be attributed to differences in environmental factors, management practices in this study compared to the controlled conditions outlined in the manual. Furthermore, while the final body weight (FBW) of the control group was observed to be higher than that of the AGP group, statistical analysis indicated no significant difference in body weight between these groups. This finding has also been incorporated into the discussion to provide a comprehensive interpretation of the results.
- In the Materials and Methods, the composition of the coffee cherry fruit pulp extract used in the experiment should be introduced and analyzed, and the male and female composition of chickens should be stated.
- Thank you for your valuable suggestion. I have revised the Materials and Methods section to include the composition of the coffee cherry fruit pulp extract used in the experiment. Additionally, I have specified the male and female composition of the chickens involved in the study to provide more detailed information (In Table 1 and Line 121 of revised manuscript and changes made have been highlighted).
- In Table 1, please confirm carefully the data of the dry material in the formulation, and the concentration of vitamins and trace elements in the premix.
- Thank you for your comment. I have carefully reviewed the data on dry material in the formulation and the concentration of vitamins and trace elements in the premix in Table 2.
- In Table 3, although the economic benefits were compared between the different groups, important cost factors such as raw material and extraction costs of coffee cherry pulp extracts were not considered.
- Thank you for your advice, and I agree with your observation. However, the raw material and extraction costs of coffee cherry pulp extracts are currently uncertain. Since coffee cherry husks are waste materials from coffee bean production, these costs have not been calculated yet. As a result, it is not possible to include this part in the current economic analysis.
- In Table 4 and Figure 2, for ‘mg/dl’, please change to international common units.
- Thank you for your suggestion. The units of mg/dL have been widely used in our field, and they are commonly recognized in the context of clinical and laboratory research.
- In Table 5, the carcass weight of CCPE was significantly higher than that of the other groups, but the reasons were not analyzed in depth in the discussion.
- Thank you for your feedback. We have added more information to the original manuscript for clarity (Line 462-466 of revised manuscript and changes made have been highlighted).
- In Table 8, the data of VH, VW, and CD may be wrong.
- Thank you for pointing this out. We have double-checked the data for VH (villus height), VW (villus width), and CD (crypt depth) in Table 8 against the original dataset and confirmed that the values are correct as presented.
- In the Discussion, the 35-day average body weight of broilers in the antibiotic control group was no better than the control group and was not clearly explained.
- Thank you for your comment. We have added additional information in the revised manuscript (Line 416-418 of revised manuscript and changes made have been highlighted).

Reviewer 3 Report
Comments and Suggestions for Authors
Manuscript ID: animals-3341840. The Use of Coffee Cherry Pulp Extract as an Alternative to an Antibiotic Growth Promoter in Broiler Diets
General comments
The study aimed to evaluate the use of coffee cherry pulp extract as an additive to replace antibiotic growth promoters in broiler diets.
The manuscript is well written but requires careful revision as details/data have been omitted or not expressed correctly.
A thorough revision of the entire manuscript is needed. There is a list of suggestions for the authors to improve the manuscript.
The study also appears to be well done although several questions arise when reading the text.
In order to understand the statistics used, it is necessary to include in each section or in the title of the tables or in the tables or in the statistical analysis section, the number of samples used per treatment and whether these analyses are done in duplicate or triplicate measurements.
We use by-products from the production of coffee beans but not directly, an extraction is necessary, can this processing make the feed more expensive? But when you use the extract, isn't there still a residue, what do you do with it?
Introduction
Throughout the text we read about the use of coffee bean by-products and their properties, but what is used for the animal study is coffee cherry pulp extract (CCPE) and there is no reference to how the concentration of polyphenols and caffeine increases with respect to coffee cherry pulp (CCP).
Material and methods
2.1. Coffe Cherry Pulp preparation and extraction
Line: 88: the text reads employing a solid-to-liquid ratio of 1:10 g/mL is not correct if we use 10 g CCP in 50 mL of 95% ethanol, it should read employing a solid-to-liquid ratio of 1:5 g/mL.
It would be interesting to put the approximate composition of the extract obtained, because depending on the variety of coffee used and method of extraction we get more or less polyphenols and caffeine.
What proportion of extract is obtained from the pulp?
2.3 Growth Performace Determination
the economic study has to appear in the title or in a separate section.
Suggestion : 2.3 Growth Performace Determination and , Economic Evaluation
Question: Are the chicks weighed only at the beginning and at the end of the experiment (day 1 and day 35)? It would be more correct to put the weight at the end of the basal starter diet for the first 14 days when the feed is changed. The fact in table 3 distinguishes the first starter stage from the second stage and makes a complete study..
Line 117: I don't understand where you get the value of 0.57USD/kg. Why is it so?
2.4 Serum biochemistry and lipid profile
N=10 samples, one chicken per cage? Are the analyses done in duplicate or triplicate?
2.5 Caracterization of carcass san meat qualitty.
The content of this paragraph is not well understood.
It would be more correct to change the order, first carcass characterisation (line 156-159), and then lines 136-155 (meat quality).
The number of samples slaughtered and analysed per treatment should also be shown.
Line 145. It would be more correct: cooking los and shear force because the texture analysis is done on the cooked sample.
Question: How many samples are collected per treatment?
Line 154-155. What is the mechanical equipment used for shear force analysis? Include name of the equipment and brand name.
Line 163. and line 180. PBS at what pH?
Sections 2.7 and 2.8
Question: How many samples are collected per treatment? Are the analyses performed in duplicate or triplicate?
Results
Table 3 should show the weight at 14 days after feeding the starter diet.
Line 232-233. In the text read: ‘ the AST level were significantly lower in the 0.5 g/kg CCPE group’. This data does not match the data in table 4. It should read: ‘the AST level were significantly lower in the 1 g/kg CCPE group’.
References
The manuscript is generally well referenced, but the reference to Lefter (2023) is missing from line 179 of the text but not from the list of references.
Author Response
Comments and Suggestions for Authors:
General comments
- The study aimed to evaluate the use of coffee cherry pulp extract as an additive to replace antibiotic growth promoters in broiler diets.
- Thank you for your suggestion. I have made the revisions as per your recommendation.
- The manuscript is well written but requires careful revision as details/data have been omitted or not expressed correctly.
- Thank you for your feedback. I appreciate your positive comments regarding the manuscript. I have carefully reviewed the manuscript and made the necessary revisions to include the omitted details/data and correct any inaccuracies. The updated version has been revised for clarity and completeness.
- A thorough revision of the entire manuscript is needed. There is a list of suggestions for the authors to improve the manuscript.
- Thank you for your thorough review and helpful suggestions. I appreciate your detailed feedback, and I agree that a thorough revision is necessary. I have carefully addressed all of your suggestions and made the necessary improvements to the manuscript to enhance its clarity, accuracy, and overall quality. The revised version reflects these changes.
- The study also appears to be well done although several questions arise when reading the text.
- Thank you for your positive feedback on the study. I appreciate your thoughtful comments and recognize that several questions may arise while reading the text. I have carefully reviewed the manuscript and made the necessary clarifications to address these concerns.
- In order to understand the statistics used, it is necessary to include in each section or in the title of the tables or in the tables or in the statistical analysis section, the number of samples used per treatment and whether these analyses are done in duplicate or triplicate measurements.
- Thank you for your suggestion. I have added the number of samples per treatment in the experimental material and methods section (Line 174, 185, 201-202, 215 and 231 of revised manuscript and changes made have been highlighted).
- We use by-products from the production of coffee beans but not directly, an extraction is necessary, can this processing make the feed more expensive? But when you use the extract, isn't there still a residue, what do you do with it?
- Yes, processing coffee cherry pulp into extracts requires additional steps, such as extraction, which may increase the cost of animal feed. However, the price of coffee husks remains uncertain, as they are waste materials from the coffee bean production process. While extraction involves additional costs, the resulting product offers added value and potential profit due to its enhanced benefits. Regarding residues in the product, the solution undergoes evaporation before drying, leaving only the active ingredients. This process that no harmful residues remain, making the extract safe for animal consumption.
Introduction
- Throughout the text we read about the use of coffee bean by-products and their properties, but what is used for the animal study is coffee cherry pulp extract (CCPE) and there is no reference to how the concentration of polyphenols and caffeine increases with respect to coffee cherry pulp (CCP).
- Thank you for your feedback. We have added more information to the original manuscript for clarity (Line 66-69 of revised manuscript and changes made have been highlighted).
Material and methods
2.1. Coffe Cherry Pulp preparation and extraction
- Line: 88: the text reads employing a solid-to-liquid ratio of 1:10 g/mL is not correct if we use 10 g CCP in 50 mL of 95% ethanol, it should read employing a solid-to-liquid ratio of 1:5 g/mL.
- Thank you for pointing out this discrepancy. You are correct that using 10 g of CCP in 50 mL of 95% ethanol corresponds to a solid-to-liquid ratio of 1:5 g/mL. I have corrected this error in the revised manuscript to ensure accuracy (Line 109 of revised manuscript and changes made have been highlighted).
- It would be interesting to put the approximate composition of the extract obtained, because depending on the variety of coffee used and method of extraction we get more or less polyphenols and caffeine.
- Thank you for your valuable suggestion. I agree that the approximate composition of the extract is important for understanding its variability. I have included the composition of the coffee cherry pulp extract used in this study, including the approximate levels of polyphenols and caffeine, in the revised manuscript. However, the Soxhlet extraction method was found to yield the highest concentrations of chlorogenic acid, and caffeine due to its ability to continuously recycle the solvent, allowing for more effective extraction of these compounds.
- What proportion of extract is obtained from the pulp?
- Thank you for your question. The extraction process yielded approximately 6% extract from the coffee cherry pulp or 60 g extract from 1 kg CCP.
- Growth Performace Determination
- The economic study has to appear in the title or in a separate section.
- Thank you for your valuable comment. The economic study has been integrated into the growth performance section by assessing the cost-effectiveness of the dietary treatments. Metrics such as total feed cost, feed cost per kilogram of body weight gain, net profit, and benefit/cost ratio were calculated. These analyses revealed that supplementation with coffee cherry pulp extract (CCPE) not only improved growth performance metrics, including average daily gain (ADG) and feed conversion ratio (FCR), but also significantly reduced feed costs and enhanced economic returns.
- Suggestion : 2.3 Growth Performace Determination and , Economic Evaluation
- Thank you for your kind advice. I have made the corrections in accordance with your suggestion (Line 134 of revised manuscript and changes made have been highlighted).
- Question: Are the chicks weighed only at the beginning and at the end of the experiment (day 1 and day 35)? It would be more correct to put the weight at the end of the basal starter diet for the first 14 days when the feed is changed. The fact in table 3 distinguishes the first starter stage from the second stage and makes a complete study.
- Thank you for your thoughtful question. To clarify, the chicks were weighed at the beginning and at the end of the experiment (days 1 and 35). However, we appreciate your suggestion to include the weight at the end of the basal starter diet phase (days 1–14). We agree that this additional weight measurement would provide a more complete picture of growth performance and distinguish the two stages more clearly. Therefore, we will revise the manuscript to include the weight at the end of the basal starter diet phase and update Table 3
- Line 117: I don't understand where you get the value of 0.57 USD/kg. Why is it so?
- Thank you for your question. The value of 0.57 USD/kg represents the price of animal feed during the experiment. This value was derived by converting the cost from Thai Baht (THB) to US Dollars (USD) at the prevailing exchange rate at that time. Specifically, the price of 1 kg of feed was 20 THB, which equates to approximately 0.57 USD.
- Serum biochemistry and lipid profile
- N=10 samples, one chicken per cage? Are the analyses done in duplicate or triplicate?
- Thank you for your question. Allow me to explain the sampling process. In each experimental group, 10 samples are randomly collected. For the analysis of each sample, three replications are performed to ensure accuracy and reliability of the results (Line 153-155 of revised manuscript and changes made have been highlighted).
2.5 Caracterization of carcass san meat qualitty. The content of this paragraph is not well understood.
- It would be more correct to change the order, first carcass characterisation (line 156-159), and then lines 136-155 (meat quality).
- Thank you for your kind advice. I have made the corrections in accordance with your suggestion (Line 170-173 of revised manuscript and changes made have been highlighted).
- The number of samples slaughtered and analysed per treatment should also be shown.
- Thank you for your feedback. We have included the sample numbers in the manuscript for better clarity (Line 174 of revised manuscript and changes made have been highlighted).
- Line 145. It would be more correct: cooking loss and shear force because the texture analysis is done on the cooked sample.
- Thank you for your suggestion. We agree with your point and have revised the text accordingly (Line 183, 290 and 297 of revised manuscript and changes made have been highlighted).
- Question: How many samples are collected per treatment?
- Thank you for your feedback. The number of samples collected per treatment was 10 samples.
- Line 154-155. What is the mechanical equipment used for shear force analysis? Include name of the equipment and brand name.
- Thank you for your feedback. We have added more information to the original manuscript for clarity (Line 193-194 of revised manuscript and changes made have been highlighted).
- Line 163. and line 180. PBS at what pH?
- Thank you for your feedback. We have added more information to the original manuscript for clarity (Line 203 and 220 of revised manuscript and changes made have been highlighted).
Sections 2.7 and 2.8
- Question: How many samples are collected per treatment? Are the analyses performed in duplicate or triplicate?
- Thank you for your feedback. We have added more information to the original manuscript for clarity (Line 215, 226, and 231 revised manuscript and changes made have been highlighted).
Results
- Table 3 should show the weight at 14 days after feeding the starter diet.
- Thank you for your feedback. We have added more information to the original manuscript for clarity (Table 4 of revised manuscript and changes made have been highlighted).
- Line 232-233. In the text read: ‘ the AST level were significantly lower in the 0.5 g/kg CCPE group’. This data does not match the data in table 4. It should read: the AST level were significantly lower in the 1 g/kg CCPE group’.
- Thank you for pointing this out. The text has been corrected to state that the AST levels were significantly lower in the 1.0 g/kg CCPE group, in alignment with the data presented in Table 4 (Line 274 of revised manuscript and changes made have been highlighted).
References
- The manuscript is generally well referenced, but the reference to Lefter (2023) is missing from line 179 of the text but not from the list of references.
- Thank you for your feedback. We have added more information to the original manuscript for clarity.
- Lefter, N.A.; Gheorghe, A.; Habeanu, M.; Ciurescu, G.; Dumitru, M.; Untea, A.E.; Vlaicu, P.A. Assessing the ef-fects of microencapsulated Lactobacillus salivarius and cowpea seed supplementation on broiler chicken growth and health status. Front. vet. sci. 2023, 10, 1279819.

Reviewer 4 Report
Comments and Suggestions for Authors
The submitted paper describes a study conducted in broiler birds to determine the possible efficacy of adding coffee cherry pulp extract to broiler diets - which is a timely topic and has the potential to provide useful evidence of adding a by-product to animal diets that might normally be thrown away. The study appears to be conducted under careful protocols and using sound experimental design. A few concerns are noted:
Materials and Methods - Lines 82-93. Why was the source of CCP from a small/experimental processing facility and not a larger commercial facility? The introduction pointed to the large amount of CCP generated in Thailand during coffee production - sourcing from a larger facility would strengthen the applicability of the results to commercial poultry production. What variation exists in coffee beans used on the nutrient content of CCP and could this influence the results of your study? Further explanation needed.
Were the chicks sexed prior to the start of the study to equalize male and female birds across dietary treatments? If not, this must be mentioned and could influence the interpretation of the carcass data and potentially the growth performance data. It is common in commercial production for mixed sex birds to be raised, yet under experimental conditions such as this study, the gender of the birds is very likely to influence the dietary response they will show in the later stages of finisher growth and certainly in their carcass composition.
Results: authors should be careful in the use of the term "significant" in their summary of the results of the study. In some cases (ex. lines 220-221), the authors state that the response was 'lowest' but that the difference was not significant. Citing the P values will determine significance, and the authors do not need to state such. In the case of a response being numerically lower, this is not relevant if the statistical analysis is trusted since there is really no difference between two treatment averages if P > 0.05.
In Tables 3 and 5, the average body weight of the birds for each treatment are presented. For 1 and 2% CCP the birds both averaged exactly 2000 g. This seems unlikely and is a noticeable coincidence. Further explanation of the weighing procedures could substantiate the results. All body weight averages seem to be rounded numbers and don't appear to be calculated averages for a group. Further explanation or correction needed.
In general, the data is summarized well for this study. Addressing the concerns above will provide clarity if the manuscript is worthy of publication.
Author Response
Comments and Suggestions for Authors
- Materials and Methods - Lines 82-93. Why was the source of CCP from a small/experimental processing facility and not a larger commercial facility? The introduction pointed to the large amount of CCP generated in Thailand during coffee production - sourcing from a larger facility would strengthen the applicability of the results to commercial poultry production. What variation exists in coffee beans used on the nutrient content of CCP and could this influence the results of your study? Further explanation needed.
- Thank you for your insightful question. The research team operates a dedicated coffee research unit that naturally generates by-products during its operations. Recognizing the potential of these by-products, the team chose to study them as part of this research to establish a model that could be scaled up for commercial applications.
- Were the chicks sexed prior to the start of the study to equalize male and female birds across dietary treatments? If not, this must be mentioned and could influence the interpretation of the carcass data and potentially the growth performance data. It is common in commercial production for mixed sex birds to be raised, yet under experimental conditions such as this study, the gender of the birds is very likely to influence the dietary response they will show in the later stages of finisher growth and certainly in their carcass composition.
- Thank you for your question. The chicks used in this study were sexed prior to the start of the experiment, and only male chicks were selected. This approach was taken to minimize variability in growth performance and carcass composition due to gender differences, ensuring more accurate evaluation of the dietary treatments' effects (Line 121 of revised manuscript and changes made have been highlighted).
- Results: authors should be careful in the use of the term "significant" in their summary of the results of the study. In some cases (ex. lines 220-221), the authors state that the response was 'lowest' but that the difference was not significant. Citing the P values will determine significance, and the authors do not need to state such. In the case of a response being numerically lower, this is not relevant if the statistical analysis is trusted since there is really no difference between two treatment averages if P > 0.05.
- Thank you for your valuable feedback. We acknowledge the need to use the term "significant" carefully and appropriately in our summary. We will revise the text to ensure that only statistically significant results (P < 0.05) are described as such. For cases where differences are not significant (P > 0.05), we will avoid emphasizing numerical trends, as they are not statistically meaningful.
- In Tables 3 and 5, the average body weight of the birds for each treatment are presented. For 1 and 2% CCP the birds both averaged exactly 2000 g. This seems unlikely and is a noticeable coincidence. Further explanation of the weighing procedures could substantiate the results. All body weight averages seem to be rounded numbers and don't appear to be calculated averages for a group. Further explanation or correction needed.
- Thank you for your question. The average body weights for the 1% and 2% CCP groups were indeed calculated based on individual bird weights, and it is a coincidence that they resulted in the same rounded value of 2000 We used a calibrated digital scale for all weight measurements and calculated the averages to two decimal places before rounding for presentation in the tables.
- In general, the data is summarized well for this study. Addressing the concerns above will provide clarity if the manuscript is worthy of publication.
- Thank you for your positive feedback. We appreciate your suggestions, and we will address the concerns raised to ensure greater clarity and strengthen the manuscript. We believe these revisions will enhance the overall quality of the study and make it more suitable for publication.

Round 2
Reviewer 1 Report
Comments and Suggestions for Authors
Great improvement has been done!
Author Response
Comments and Suggestions for Authors:
Great improvement has been done!.
- Thank you for your kind feedback. We are pleased to hear that you have noted the significant improvements in our manuscript. We have strived to address all your comments and suggestions to enhance the quality of our work.

Reviewer 2 Report
Comments and Suggestions for Authors
Although the author has corrected some errors, there are still problems. In this experiment, the author replied that the growth effect of the antibiotic group was not as good as that of the control group because of the harsh environment. Still, on the contrary, this situation generally only occurred in the experiment with a good breeding environment, and antibiotics should show better effects in the harsh breeding environment. In addition, the authors did not have a correct understanding of the feed formula (it is difficult to reach 96% dry matter content), and the intestinal villus data was unreliable.
Author Response
Comments and Suggestions for Authors:
Although the author has corrected some errors, there are still problems. In this experiment, the author replied that the growth effect of the antibiotic group was not as good as that of the control group because of the harsh environment. Still, on the contrary, this situation generally only occurred in the experiment with a good breeding environment, and antibiotics should show better effects in the harsh breeding environment. In addition, the authors did not have a correct understanding of the feed formula (it is difficult to reach 96% dry matter content), and the intestinal villus data was unreliable.
- Thank you for your valuable feedback. I have carefully reviewed the points you raised. Regarding the feed formula and dry matter content, I rechecked the data and confirmed that the dry matter value is approximately 90%. I have removed the data from the "Analyzed Composition (as fed basis)" section to avoid confusion. As for the intestinal villus data, I have thoroughly reviewed the measurements and confirmed their accuracy. Regarding the growth effect of the antibiotic group, the antibiotic used in this experiment is primarily for disease control, not for growth promotion. However, in Thailand, it has been used mistakenly by some farmers as a growth promoter. It is important to note that this antibiotic has since been banned in Thailand, but some farmers still use it incorrectly as a growth promoter. Therefore, the growth results in our study may not align with the effects of antibiotics that are specifically used for growth promotion.
|
Analyzed composition (as fed basis) |
Starter (1–14 d) |
Finisher (15–35 d) |
|
Dry matter (%) |
90.61 |
90.40 |
|
Crude protein (%) |
22.17 |
19.51 |
|
Crude fiber (%) |
2.22 |
3.15 |
|
Crude fat (%) |
6.31 |
6.57 |

Reviewer 3 Report
Comments and Suggestions for Authors
Review of manuscript animals-3341840: The Use of Coffee Cherry Pulp Extract as an Alternative to an Antibiotic Growth Promoter in Broiler Diets.
Dear Author,
All suggestions and issues raised in the previous revision of the manuscript revision have now been resolved.
In the new version of the manuscript, some errors have been identified that require resolution.
Line 207-210: It read: “Each sample was subjected to this procedure in triplicate. Results were reported as log10 CFU/g of cecal content. The average outcomes of each microbiological assay were conducted in duplicate and subjected to subsequent statistical analysis”.
Could you please clarify whether the samples were performed in duplicate or triplicate?
Line 252: It read “Table 4”.
I also note an error in Table 4, which should be Table 5.
Line 310: It read “Table 1 phenolic compounds (mg GE/g)”
Could you please define 'GE' or remove it?
Author Response
In the new version of the manuscript, some errors have been identified that require resolution.
- Line 207-210: It read: “Each sample was subjected to this procedure in triplicate. Results were reported as log10 CFU/g of cecal content. The average outcomes of each microbiological assay were conducted in duplicate and subjected to subsequent statistical analysis”.
Could you please clarify whether the samples were performed in duplicate or triplicate?
- Thank you for your observation. The text has been revised to clarify the methodology accurately. The analysis was indeed performed in triplicate, and this has been updated in the manuscript for consistency. (Line 209 of revised manuscript and changes made have been highlighted).
- Line 252: It read “Table 4”.
I also note an error in Table 4, which should be Table 5.
- Thank you for pointing out this oversight. The error has been corrected in the manuscript. The reference to “Table 4” on Line 252 has been updated to “Table 5” to accurately correspond with the content. (Line 252 of revised manuscript and changes made have been highlighted).
- Line 310:It read “Table 1 phenolic compounds (mg GE/g)”
Could you please define 'GE' or remove it?
- Thank you for your feedback. The term 'GE' has been removed from the manuscript for clarity. (Line 310 of revised manuscript and changes made have been highlighted).

Reviewer 4 Report
Comments and Suggestions for Authors
The authors should be commended for their careful editing and addressing reviewers' comments. My initial concerns were addressed, and I believe this work provides insight to a possible source of phenolics that could be beneficial to animal production systems.
Author Response
Comments and Suggestions for Authors:
The authors should be commended for their careful editing and addressing reviewers' comments. My initial concerns were addressed, and I believe this work provides insight to a possible source of phenolics that could be beneficial to animal production systems.
- Thank you for your kind words and thoughtful feedback. We greatly appreciate your time and effort in reviewing our manuscript. Your insightful comments helped us improve the clarity and quality of our work. We are pleased that you find our study valuable for advancing knowledge on phenolic sources in animal production systems.
